# The Relationship between Iron and LRRK2 in a 6-OHDA-Induced Parkinson’s Disease Model

**DOI:** 10.3390/ijms24043709

**Published:** 2023-02-13

**Authors:** Ruru Jia, Yanling Liu, Ke Shuai, Cheng Zhou, Lei Chen, Li Zhu, Xiao-Mei Wu

**Affiliations:** Laboratory of Hypoxic Biomedicine, Institute of Special Environmental Medicine, Co-Innovation Center of Neuroregeneration, Nantong University, 9 Seyuan Road, Nantong 226019, China

**Keywords:** iron, LRRK2, dopaminergic neuron, apoptosis, Parkinson’s disease

## Abstract

The pathogenesis of Parkinson’s disease (PD) is very complex and still needs further exploration. Leucine-rich repeat kinase 2 (LRRK2) is associated with familial PD in mutant forms and sporadic PD in the wild-type form. Abnormal iron accumulation is found in the substantia nigra of PD patients, but its exact effects are not very clear. Here, we show that iron dextran exacerbates the neurological deficit and loss of dopaminergic neurons in 6-OHDA lesioned rats. 6-OHDA and ferric ammonium citrate (FAC) significantly increase the activity of LRRK2 as reflected by the phosphorylation of LRRK2, at S935 and S1292 sites. 6-OHDA-induced LRRK2 phosphorylation is attenuated by the iron chelator deferoxamine, especially at the S1292 site. 6-OHDA and FAC markedly induce the expression of pro-apoptotic molecules and the production of ROS by activating LRRK2. Furthermore, G2019S-LRRK2 with high kinase activity showed the strongest absorptive capacity for ferrous iron and the highest intracellular iron content among WT-LRRK2, G2019S-LRRK2, and kinase-inactive D2017A-LRRK2 groups. Taken together, our results demonstrate that iron promotes the activation of LRRK2, and active LRRK2 accelerates ferrous iron uptake, suggesting that there exists an interplay between iron and LRRK2 in dopaminergic neurons, providing a new perspective to uncover the underlying mechanisms of PD occurrence.

## 1. Introduction

Parkinson’s disease (PD) is the second most common neurodegenerative disorder and is characterized by the progressive loss of dopaminergic neurons in the substantia nigra of patients. Accumulated evidence has demonstrated that PD is caused by multiple factors, including genetic mutations and environmental toxins [1,2]. Although familial PD with a low incidence rate has been revealed to be associated with mutations in related genes such as *SNCA* and *LRRK2* [3,4], the etiopathogenesis of sporadic cases with higher incidence rate remains unclear. Recently, it was reported that besides age and environmental factors, the wild-type familial PD genes are also involved in sporadic disease by influencing the susceptibility to PD [5,6]. Environmental toxicants such as rotenone, paraquat, and manganese induce pathogenic post-translational modification of α-synuclein and leucine-rich repeat kinase 2 (LRRK2), increasing the risk of PD [7]. These may open new windows for elucidating the mechanisms underlying the occurrence of PD.

LRRK2, also known as Dardarin, is encoded by a large gene, *LRRK2*, which has 2527 amino acids and several potential protein-protein interaction regions (ARM, ANK, LRR, and WD40) surrounding a central catalytic core with GTPase and kinase activities (ROC and MAPK domains) [7]. Structural characteristics of LRRK2 suggest that it may serve as a scaffold for the assembly of protein complexes and be a central integrator of multiple signaling pathways [7]. In physiological conditions, LRRK2 plays an important role in the nervous system; for example, the wild-type LRRK2 can regulate neurite outgrowth in developing neurons by interacting with disheveled proteins or tubulins [8]. Given the current knowledge that LRRK2 plays a role cross the entire PD disease spectrum, seven mutations in LRRK2 are considered pathogenic with an autosomal dominant form of PD [9]. Pathogenic mutations cause highly significant alteration in kinase activity of LRRK2, including the most common mutation of PD, G2019S, which consistently induces 2 to 3-fold increases in kinase activity, resulting in neuronal toxicity [10]. Recent research indicates that wild-type LRRK2 is also involved in idiopathic PD, as shown by the finding that LRRK2 activity in nigrostriatal dopamine neurons was greater in individuals with the disease than in controls [11]. LRRK2 can be phosphorylated at 103 predicted amino acid residues, including constitutive and inducible phosphorylation sites. It is constitutively phosphorylated at S935, among other sites, and has been used extensively as an indirect marker of activity in cells and tissue. The dephosphorylation at S935 was validated as a pharmacodynamic biomarker in PD patients and is currently used as a readout verifying the effectiveness of LRRK2 inhibitors [7,11]. The autophosphorylation at the S1292 site has been proposed as a direct marker of LRRK2 activation since it reflects the activity of LRRK2 per se. In idiopathic patients, the level of LRRK2 phosphorylation at S1292 was approximately six times higher than in healthy controls [12]. Several PD-linked mutations, such as the G2019S mutant, enhance this phosphorylation, whereas kinase-inactive LRRK2 is not phosphorylated at this residue. In addition, some inhibitors of LRRK2 have been shown to induce S1292 dephosphorylation in vivo and in vitro studies [13,14,15]. However, it is not very clear what factors cause the increase in kinase activity of LRRK2 during the PD onset process.

It has been demonstrated that there was a significant elevation in the content of iron in the substantia nigra of PD patients [16]. Some studies believe that iron deposition has been implicated in the pathogenesis of PD, but the question of whether iron deposition represents the initial inducement or final consequence of substantia nigra degeneration remains unsolved. It is also not known whether iron is involved in the activation of LRRK2 itself or its pathway. Here, we observed for the first time the effect of iron on LRRK2 activity and the effect of LRRK2 on intracellular iron content, providing direct evidence that there is a potential link between iron and LRRK2 in SH-SY5Y cells.

## 2. Results

### 2.1. Iron Dextran Exacerbates Loss of Dopaminergic Neurons in Substantia Nigra of a 6-OHDA-Induced Rat Model of Parkinson’s Disease

We performed a unilateral injection of 6-OHDA into the right striatum of rats to produce the retrograde injury of dopaminergic neurons in the substantia nigra pars compacta. To investigate the effect of iron on the occurrence of Parkinson’s disease (PD), iron dextran was given stereotactically into the right medial forebrain bundle (MFB) of rats after 6-OHDA was injected. First, the neurological deficit of rats was observed for 30 min by apomorphine-induced rotation once a week from the 14th day post-lesioning. It was found that the number of apomorphine-induced rotations was about 7 turns/min in rats treated with 6-OHDA alone (6-OHDA) and 8 turns/min in rats treated 6-OHDA and iron dextran (6-OHDA+ Iron-dextran) on the 14th day, indicating that the PD model in rats was successfully established in our system. Then, the number of rotations gradually increased with the prolongation of time after 6-OHDA action. There were significantly more rotations in rats treated with a combination of 6-OHDA and iron-dextran than in rats treated with 6-OHDA alone on the 28th day, as measured by about 20 turns/min in the 6-OHDA+ Iron-dextran group and 16 turns/min in 6-OHDA alone group (Figure 1A). Then, the survival of dopaminergic neurons was tested by immunostaining using an anti-tyrosine hydroxylase (TH) antibody. Our results showed that the number of TH-positive neurons (TH^+^) in the substantia nigra was significantly decreased on the 28th day after 6-OHDA treatment. Moreover, they were further reduced in the combined administration group of 6-OHDA and iron-dextran compared with the 6-OHDA group (Figure 1B,C). These results suggest that iron may exacerbate the loss of dopaminergic neurons in the substantia nigra in rat models of PD induced by 6-OHDA.

### 2.2. 6-OHDA and Ferric Ammonium Citrate Induce LRRK2 Activation and a Decrease in Viability of SH-SY5Y Cells

It has been demonstrated that there was an evident accumulation of iron in the substantia nigra of PD patients, and 6-OHDA was one of the most frequently used toxins in models of PD. First, SH-SY5Y cells were exposed to different concentrations of 6-OHDA or ferric ammonium citrate (FAC, #F5879, Sigma, Burlington, MA, USA) for 24 h. The MTT assay showed that 6-OHDA caused damage to SH-SY5Y cells in a dose-dependent manner, as reflected by the decreased cell viability to 90% (25 μM), 79% (50 μM), and 61% (100 μM) of control, in which 100 μM of 6-OHDA was chosen to use in subsequent experiments. Using different concentrations of FAC for 24 h, cell viability gradually decreased with increasing FAC concentration, reaching about 86% (2.5 μg/mL), 69% (5 μg/mL), and 49% (10 μg/mL) of the control, in which 5 μg/mL of FAC was appropriate for subsequent experiments (Appendix A). When SH-SY5Y cells were treated with 100 μM of 6-OHDA or 5 μg/mL of FAC for 24 h simultaneously, the cell viability showed a similar reduction compared with the control (Figure 1A). This observation was also consistent with the results obtained from the LDH assay in SH-SY5Y cells, which showed that LDH leakage induced by 6-OHDA was 206% of the control and by FAC was 163% of the control (Figure 1B).

To investigate the underlying mechanisms by which iron promotes the occurrence of PD, we analyzed the activity of LRRK2, which is likely central in the pathogenesis of PD, in SH-SY5Y cells treated with 6-OHDA and FAC. Since Ser1292 autophosphorylation is an indicator of LRRK2 kinase activity and pharmacological inhibition of its kinase activity can abolish Ser935 phosphorylation, the phosphorylation of LRRK2 at Ser935 and Ser1292 were detected by Western blot. It was found that 6-OHDA and FAC both markedly induced the LRRK2 phosphorylation at the Ser935 site (pS935-LRRK2), which was 2.26-fold (6-OHDA) and 1.84-fold (FAC) of control, and at the Ser1292 site (pS1292-LRRK2), which reached 1.74-fold (6-OHDA) and 1.82-fold (FAC) of control (Figure 2C,D). These results strongly imply that LRRK2 can be activated not only by 6-OHDA but also by FAC in SH-SY5Y cells.

### 2.3. Desferrioxamine Attenuates LRRK2 Activation and Cell Damage Induced by 6-OHDA

To determine whether iron was involved in 6-OHDA-induced activation of LRRK2, the iron chelator desferrioxamine (DFO) was used to treat SH-SY5Y cells with or without 6-OHDA for 24 h. The MTT assay showed that DFO at 10, 20, and 40 µM had no effect on the cell viability of SH-SY5Y cells without 6-OHDA except at 60 µM (Figure 3A). The LDH assay showed that DFO significantly reduced the LDH leakage from cells with 6-OHDA treatment in a dose-dependent manner (Figure 3B), implying that DFO may prevent SH-SY5Y cells from developing a 6-OHDA-induced lesion. Interestingly, the elevated level of pS935-LRRK2 induced by 6-OHDA was inhibited by 40 µM of DFO (from 2.17 to 0.89-fold of control, Figure 3C). More importantly, the level of pS1292-LRRK2 was also remarkably decreased by 10 µM DFO (from 1.73 to 0.55-fold) and 40 µM DFO (from 1.73 to 0.37-fold of control, Figure 3D) in the presence of 6-OHDA. These data suggest that 6-OHDA-induced activation of LRRK2 is mediated by iron.

### 2.4. Inhibitor of LRRK2 Prevents Phosphorylation of LRRK2, Expression of Pro-Apoptotic Molecules, and Production of ROS Induced by 6-OHDA and Ferric Ammonium Citrate

A LRRK2 G2019S mutation has been reported to activate an apoptotic pathway, causing the degeneration of dopaminergic neurons in a transgenic mouse model of PD. Therefore, we tried to determine whether LRRK2-dependent signal transduction pathways play a role in the deleterious effects of iron or 6-OHDA on the dopaminergic cells. SH-SY5Y cells were pre-incubated with LRRK2-IN-1 (#438193, Calbiochem) [17], a specific inhibitor on LRRK2 activation, at a concentration of 1 or 3 µM for 1 h before treatment with 6-OHDA and FAC. Several sets of evidence suggest that activated LRRK2 induced by 6-OHDA and FAC initiates the pro-apoptotic pathway.

First, we carried out the Western blot method to examine the levels of LRRK2 phosphorylation using the special anti-phosphorylated LRRK2 at S935 or S1292 antibody. It was found that combined administration of 6-OHDA and FAC (6-OHDA+FAC) promoted the phosphorylation of LRRK2 at S935 and S1292, which was markedly inhibited by LRRK2-IN-1, especially at 3 µM of concentration (Figure 4A,B). These results imply that the activity of LRRK2 can be reflected by the levels of pS935-LRRK2 and pS1292-LRRK2, and LRRK2-IN-1 may block the induction of 6-OHDA and FAC on LRRK2 activation. Second, we asked whether Bim and FasL, the downstream effectors of LRRK2, were involved in the process of lesion induction by 6-OHDA or FAC. RT-PCR results showed that the levels of Bim and FasL mRNA were dramatically raised under 6-OHDA, FAC, and 6-OHDA+FAC conditions, which were also inhibited by LRRK2-IN-1 compared to their respective reference groups (Figure 4C,D), suggesting that 6-OHDA and FAC induced the expression of pro-apoptotic molecules via LRRK2.

It has been reported that LRRK2 regulates ROS production in both SH-SY5Y and RAW cell lines [18]. Finally, to determine whether 6-OHDA and FAC caused ROS production via LRRK2, LRRK2-IN-1 (IN-1) was used to act on SH-SY5Y cells before treatment with 6-OHDA+FAC. The DCFH-DA assay showed that 6-OHDA+FAC dramatically induced ROS production (344% of control), which was markedly decreased by LRRK2-IN-1 (163% of control, Figure 4G). LRRK2-IN-1 had a similar effect to N-acetylcysteine (NAC), which is an antioxidant that scavenges ROS (from 339to 174% of control) in SH-SY5Y cells. These data demonstrate that LRRK2 exerts thane important role in the process of ROS generation caused by 6-OHDA and FAC in SH-SY5Y cells.

### 2.5. Activated LRRK2 Increases Total Iron Content and Ferrous Iron Uptake in SH-SY5Y Cells

Our present study has already demonstrated that iron could promote the activation of LRRK2 in SH-SY5Y cells. But in turn, whether there is an effect of activated LRRK2 on iron homeostasis was little known. The high nigral iron deposition was found in LRRK2 and Parkin mutation carriers using MRI R2* relaxometry [19]. The LRRK2 G2019S mutation is the most common causative genetic factor linked to PD, which resulted in increased in-vitro kinase activity. We observed whether there was an effect of hyperactive LRRK2 on the level of iron itself in SH-SY5Y cell lines stably expressing wild-type LRRK2 (WT-LRRK2), the G2019S mutant (G2019S-LRRK2), and the kinase-inactive mutant D2017A (D2017A-LRRK2). The intracellular total iron content was examined by reacting with Ferene S after ferric reduction to the ferrous form. It was found that the total iron content was remarkably increased in SH-SY5Y cells stably expressing G2019S-LRRK2 compared with cells transfected with an empty vector and cells expressing WT-LRRK2, but not seen in cells expressing D2017A-LRRK2 (Figure 5A). Next, the ferrous iron uptake was investigated by the Fe^2+^-sensitive fluorescent dye Calcein-AM, which could be quenched upon binding ferrous iron. Without addition of ferrous ammonium sulfate (FAS), the fluorescence remained steady during the 30-min recording period, but it dramatically decreased after FAS was added, indicating that a large amount of ferrous iron was taken into cells. Compared to control cells or cells containing empty vectors, the uptake onf FAS was further increased in cells stably expressing WT-LRRK2, G2019S-LRRK2, and D2017A-LRRK2. Interestingly, among these three groups, G2019S-LRRK2 mutant cells showed the strongest absorption capacity for ferrous iron (Figure 5B). Taken together, our results indicate that the active LRRK2 can increase intracellular iron content through enhancing ferrous iron uptake in dopaminergic neurons.

## 3. Discussion

Elevated levels of iron have been found in the substantia nigra of PD patients [19], thus speculating that there is an association between the accumulation of iron and the pathogenesis of PD. However, whether iron is the causative factor of PD remains controversial, and one of the major objectives of this study was to investigate this issue. Our study presents for the first time that iron dextran exacerbated the neurological deficit and loss of dopaminergic neurons in substantia nigra of 6-OHDA lesioned rats (Figure 1). 6-OHDA increased the release of LDH in SH-SY5Y cells, which was blocked by the iron chelator DFO in a dose-dependent manner (Figure 3B). These two results suggest that iron may be involved in the degeneration of dopaminergic neurons and that 6-OHDA-caused cellular injury is achieved by iron. LRRK2 causes not only familial PD in pathogenic mutations but is also involved in idiopathic PD in its wild-type form because of its high activity [11]. However, it is not yet clear whether iron can affect LRRK2 activity or whether the reverse exists. By observing the change of LRRK2 phosphorylation (Figure 2, Figure 3 and Figure 4A,B) and iron uptake (Figure 5B,C) in SH-SY5Y cells, we provide the direct evidence necessary to answer our hypothesis. Our results establish an interaction in theory between iron and LRRK2 activity: iron may promote LRRK2 activation, and active LRRK2 accelerates iron uptake. This offers significant advances in revealing the underlying mechanisms of PD occurrence, as detailed below.

One important novel finding of the present study is that 6-OHDA and FAC both could significantly induce the phosphorylation of LRRK2 at S935 and S1292 of SH-SY5Y cells. It has been confirmed that PD-related mutations in LRRK2 enhance the kinase activity of the protein [13]. LRRK2 phosphorylation activity on S910, S935, and S1292 is linked with the regulation of its kinase activity, especially in autophosphorylation of S1292 which is a direct indicator of LRRK2 kinase activity. Thus, measuring the phosphorylation at S935 and S1292 can reflect the activation of LRRK2, which are very useful biomarkers to monitor its kinase activity. Our results demonstrate that, in addition to 6-OHDA, FAC induces phosphorylation of LRRK2 in SH-SY5Y cells (Figure 2C,D), suggesting that 6-OHDA and FAC both can elevate the activity of LRRK2, and FAC has the same effect on LRRK2 activation as 6-OHDA. In future studies, effective commercial antibodies against phospho-LRRK2 are urgently needed to investigate the level and activation of LRRK2 in animal brains. Interestingly, we found that 6-OHDA-induced phosphorylation of LRRK2 at S935 was attenuated in the presence of 40 µM DFO (Figure 3C), and the phosphorylation of S1292 also was significantly inhibited by 10 or 40 µM of DFO, in which the phosphorylation of S1292 showed more susceptibility to DFO than that of S935 (Figure 3D). These data imply that the activation of LRRK2 induced by 6-OHDA is mediated by iron, and iron may play a critical role in the activation of LRRK2 in dopaminergic neurons.

Our experiments also unveil the potential mechanisms by which iron mediates cellular injury through activation of LRRK2. LRRK2, as an upstream regulator in events leading to neurodegeneration, is a very large multi-domain protein harboring both GTPase and kinase activity. It has cytotoxic catalytic activity and possesses a main function linked to the assembly of signaling complexes [7,11,20]. LRRK2 binds to three MAPKKs (MKK3, MKK6, and MKK7) and four MAPK scaffolding proteins (JNK-interacting proteins 1–4), by which p38 MAPK and JNK are activated to facilitate cellular apoptosis in animal PD models. For example, it plays a critical role in neuronal death by directly phosphorylating and activating apoptosis signal-regulating kinase 1 (ASK1) and acts as a scaffolding protein by interacting with each component of the ASK1–MKK3/6–p38 MAPK pathway [21,22,23]. In the present study, it was observed that combined administration with 6-OHDA and FAC significantly increased the levels of pS935-LRRK2 and pS1292-LRRK2, which were inhibited by LRRK2-IN-1, a specific inhibitor of LRRK2 (Figure 4A and 4B), indicating that LRRK2-IN-1 could be used to study the kinase activity and that it is an effective inhibitor of 6-OHDA and FAC-induced LRRK2 activation in our experiment system. Subsequently, we found that 6-OHDA and FAC markedly induced the expression of Bim as well as FasL, which were significantly decreased when cells were pre-treated with LRRK2-IN-1 (Figure 4C-D), implying that LRRK2 played a pivotal role in the process by which 6-OHDA and FAC caused the expression of pro-apoptotic factors.

It has been reported that H_2_O_2_ increased LRRK2 kinase activity in mouse primary cortical neurons, and a sub-toxic dose of H_2_O_2_ augmented LRRK2-induced dopaminergic neuron loss [24]. Conversely, LRRK2 affects mitochondrial function, leading to exacerbation of ROS generation and susceptibility to oxidative stress [18]. The G2385R-LRRK2 mutant also increases mitochondrial ROS and activates caspase-3/7, resulting in neurotoxicity [25]. Here, our study found that combined treatment of 6-OHDA and FAC markedly raised the level of ROS in SH-SY5Y cells; however, this increase was significantly reduced by LRRK2-IN-1, which was similar to that of an antioxidant NAC (Figure 4E,F), suggesting that despite the fact that 6-OHDA and FAC themselves could produce ROS in cells, the production of ROS might be mainly mediated by LRRK2. These results imply that 6-OHDA and FAC could cause insults to dopaminergic cells by activating LRRK2 to generate ROS.

Finally, we have demonstrated that iron could facilitate the activation of LRRK2 in SH-SY5Y cells; on the contrary, whether active LRRK2 regulated iron homeostasis was little known. High iron deposition was observed in the substantia nigra of LRRK2 mutation patients [19], suggesting there exists a potential link between LRRK2 and iron. The G2019S-LRRK2 mutation is the most common causative genetic factor linked to PD because of abnormally elevated kinase activity [12,26]. Recently, it was reported that a G2019S knock-in significantly increased the iron deposition in proinflammatory conditions, focusing on microglia but not neurons or other types of cells and indirectly describing the effect of LRRK2 on iron absorption [27,28]. Our study provides direct evidence that overexpression of LRRK2 increased cellular uptake of FAS; moreover, G2019S-LRRK2 mutant cells with high kinase activity showed the strongest absorption capacity for ferrous iron among the WT-LRRK2, G2019S-LRRK2, and D2017A-LRRK2 groups (Figure 5C). Consequently, the total iron content was the highest in SH-SY5Y cells stably expressing G2019S-LRRK2 among the three groups but not seen in cells expressing the kinase-inactive D2017A-LRRK2 mutant (Figure 5B). Thus, these results suggest that the active LRRK2 may raise the content of total iron by enhancing the ferrous iron uptake in dopaminergic neurons.

In conclusion, the findings that there exists an interplay between iron and LRRK2 activation in dopaminergic cells in this study are significant in the treatment of PD in two aspects. First, we highlight the importance of iron as a key mediator promoting the activation of LRRK2, which induces the expression of pro-apoptotic molecules and the production of ROS. Second, abnormally activated LRRK2 could enhance ferrous iron uptake, leading to an increase in the level of total iron. However, due to the very low purity of dopaminergic neurons in primary cultures and the lack of effective commercial anti-bodies for rodent pLRRK2/LRRK2, the findings in this study were not detected in animals to observe the effect of iron on LRRK2 activity, which is needed for further study in the future. In fact, one important implication of the present study is that a combined approach that targets both iron and LRRK2 will likely achieve a higher therapeutic efficacy. Future studies that aim to achieve this objective are worth pursuing.

## 4. Materials and Methods

### 4.1. 6-OHD- Induced PD Model and Iron Dextran Administration

The animal study was reviewed and approved by the Ethics Committee of Nantong University (S20220224-001) and conducted according to the Nantong University IACUC-approved protocol. Male Sprague-Dawley rats (3 months old, weighing 220–240 g) were provided by the Experimental Animal Center of Nantong University (Jiangsu, China). For 6-OHDA lesions, rats were anesthetized with tribromoethanol (300 mg/kg, i.p.) and immobilized in a stereotaxic apparatus (Stoelting Instruments, Wood Dale, IL, USA). 20 μg of 6-Hydroxydopamine hydrobromide (6-OHDA, #H116, Sigma) was dissolved in 4 μL of sterile saline (5 μg/μL) containing 0.02% ascorbic acid and kept on ice protected from light. This was unilaterally injected into two sites of the right striatum with the following coordinates: (1) anteroposterior (AP), 1.0 mm; mediolateral (ML), −3.0 mm; dorsoventral (DV), 4.5 mm and (2) AP, 1.0 mm; ML, −3.0 mm; DV, 6.0 mm [29,30]. 2 μL of 6-OHDA was microinjected into each site at a rate of 0.5 μL/min. The microinjection needle was left in place for a further 10 min before being slowly withdrawn. For behavior testing, apomorphine-induced (0.5 mg/kg, s.c.) rotation was tested two weeks after 6-OHDA injection. Only those rats showing at least 7 turns per min were considered successfully induced. Then, this rotational behavior was monitored once per week, i.e., three to four weeks after 6-OHDA administration.

Iron dextran was given into the right medial forebrain bundle (MFB) of rats after 6-OHDA injection. 1 mg of iron dextran (100 μg/μL, 10 μL, #D8517, Sigma) was microinjected at a rate of 0.5 μL/min into the site at the following coordinates: AP, −4.8 mm; ML, −1.0 mm; DV, 8.1 mm. The needle was left in place for 10 min before being slowly retracted.

### 4.2. Immunohistochemistry

The brain slices were processed and assessed by immunofluorescence staining as previously described with some modifications. At the end of behavioral testing, rats were anesthetized and transcardially perfused with normal saline solution, followed by 4% paraformaldehyde. The brains were removed and post-fixed in 4% paraformaldehyde for 4 h, immersed in 20% sucrose overnight, and then transferred into a 30% sucrose solution until they sank to the bottom of the container. Coronal sections containing substantial nigra (20 μm) were made using a Leica CM3050S cryostat (Leica Microsystems, Wetzlar, Germany). The brain slices were blocked with 3% goat serum (Thermo Fisher Scientific) containing 0.3% Triton X-100 in 0.01 M PBS for 1 h and incubated with anti-tyrosine hydroxylase primary antibody (1:1000, #T8700, Sigma) at 4 °C overnight. After rinsing with PBS, the slices were incubated with Alexa Fluor 488-conjugated goat anti-rabbit IgG (H+L) for 1 h. Fluorescent images were captured by a Leica SP8 confocal imaging system (Leica, Wetzlar, Germany).

### 4.3. Measurement of Cell Viability

The SH-SY5Y cell line was obtained from Procell Life Science and Technology Co., Ltd. (Wuhan, China). Cell viability was assessed using the MTT (3-(4,5-dimethylthiazol-2-yl)-2,5-diphenyl tetrazolium bromide) assay or lactate dehydrogenase (LDH) leakage as described previously [31]. In the MTT assay, the yellow substrate MTT (#CT01-5, Sigma) was reduced to a purple formazan by succinate dehydrogenase in the mitochondrion of living cells. Briefly, SH-SY5Y cells grown in 96-well plates were incubated with 100 μL of 1% MTT for 4 h at 37^o^C, the reaction was stopped by adding 100 μL of cell lysis buffer (20% sodium dodecyl sulfate in 50% N′N-dimethylformamide) to each well, and the plate was further incubated at 37^o^C for 20 h. Finally, the absorbance was measured at the 570 nm wavelength using a microplate assay reader (Synergy 2, BioTek, USA). LDH is an intracellular enzyme that leaks into the culture medium when cell membranes are damaged. LDH activity in cellular medium was determined by a commercial LDH Kit according to the manufacturer’s instructions (#A020-2, Nanjiang Jiancheng, Nanjiang, China). LDH activity in medium was calculated and converted as a percentage of the control level.

### 4.4. qPCR Assay

Total RNA was isolated from cells using Trizol Reagent (#15596018, Invitrogen, Carlsbad, CA, USA). The first-strand cDNA was synthesized according to the manufacturer’s instructions using the RevertAid^TM^ First Strand cDNA Synthesis Kit (#K1622, Thermo Fisher Scientific, Waltham, MA, USA). The following specific primers were used in real-time RT-PCR. For Bim: forward, 5′-AGTGGGTATTTCTCTTTTGACACAG-3′; reverse, 5′-GTCTCCAATACGCCGCAACT-3′. For FasL: forward, 5′-CCAGCTTGCCTCCTCTTGAG-3′; reverse, 5′-TCCTGTAGAGGCTGAGGTGTCA-3′. For β-actin: forward, 5′-CATGTACGTTGCTATCCAGGC-3′; reverse, 5′-CTCCTTAATGTCACGCACGAT-3′. The housekeeping gene β-actin was used as an internal standard for RNA preparations and reverse transcriptase reactions. Real-time quantitative PCR was carried out in the LightCycler 96 System (Roche Diagnostics, Indianapolis, IN, USA) using Power SYBR Green PCR Master Mix (#4367659, Thermo Fisher Scientific). Each reaction contained 10 µL of 2×Power SYBR Green Master Mix, forward and reverse primers, each at a final concentration of 250 nM and 2 µL cDNA. The standard cycling conditions were 95 °C for 10 min, followed by 40 cycles at 95 °C (10 s), 60 °C (15 s), and 72 °C (20 s). All reactions were performed in triplicate. Data collection occurred during the 72 °C extension step. The qPCR data were analyzed using 2^−ΔΔCT^ methodology.

### 4.5. Western Blot Analysis

SH-SY5Y cells received different treatments; they were washed with cold PBS and lyzed in cold RIPA buffer supplemented with a cocktail of protease inhibitors. Aliquots of the cell extract containing 20 μg of protein were separated on SDS-polyacrylamide gels and transferred to PVDF membranes. The membranes were blocked with 5% non-fat milk in TBST buffer for 1 h and incubated overnight at 4 °C with one of the diluted following primary antibodies: anti-pS935-LRRK2 and anti-pS1292-LRRK2 (1:1000, #ab133450 and #ab203181, Abcam, Cambridge, UK), anti-LRRK2 (1:1000, #ab133474, Abcam), and anti-β-actin (1:10000, #A2228, Sigma). After three washes with TBST, the membrane was incubated with goat anti-mouse or anti-rabbit IRDye 800 CW IgG (1:10,000, #926-32210, #926-32211, Li-Cor) for 1 h at room temperature. The intensity of the specific bands was detected and analyzed by the Odyssey infrared imaging system at a resolution of 169 μm (Li-Cor Biosciences, Lincoln, NE, USA).

### 4.6. Measurement of Reactive Oxygen Species (ROS)

ROS were measured by a fluorometric assay with 2′, 7′-dichlorofluorescein diacetate (DCFH-DA) as previously described. DCFH-DA is a stable, nonpolar compound that readily diffuses into cells and is hydrolyzed to 2′, 7′-dichlorofluorescin (DCFH). DCFH can be oxidized by ROS to the highly fluorescent 2′, 7′-dichlorofluorescein (DCF). The fluorescence intensity of DCF is proportional to the amount of ROS in cell [31]. SH-SY5Y cells treated with 6-OHDA+FAC were washed with PBS three times and incubated with 5 μM DCFH-DA (#35845, Sigma) at 37 °C for 30 min. DCF fluorescence was quantified by the Synergy 2 Multi-Detection Microplate Reader (BioTek, Winooski, VT, USA) with an excitation/emission wavelength of 485/515 nm in 96-well fluorescent plates. DCF values were obtained after subtracting the background fluorescence levels.

### 4.7. Total Iron Content Assay

The intracellular total iron content was analyzed using the kit as described [32]. The ferric carrier protein dissociated ferric into solution in the presence of an acid buffer. After reduction to the ferrous form (Fe^2+^), iron reacted with Ferene S (an iron chromogen) to produce a stable colored complex. Levels of intracellular total iron were analyzed in the same number of control, pcDNA3.1(+), LRRK2, G2019S, and D2017A mutant SH-SY5Y cells according to the manufacturer’s instructions of the Iron Assay Kit (#ab83366, Abcam). Absorbance at the 593 nm wavelength was detected by spectrophotometry. The level of intracellular iron was calculated and converted to the fold of the control level.

### 4.8. Ferrous Iron Uptake Assay

The ferrous uptake assay was performed as described [33]. Briefly, the cells were washed twice and incubated with Calcein-AM (0.125 µM final concentration, #C1430, Thermo Fisher Scientific) in Hank’s balanced salt solution (HBSS, pH 7.4) for 10 min at 37 °C. Excess Calcein-AM was washed out three times with HBSS. Prior to measurement, 100 µL of Calcein-loaded cell suspension and 2 mL HEPES were added to the cuvette. The fluorescence intensity was measured at 488 nm excitation wavelength and 525 nm emission wavelength by ultraviolet spectrophotometer (Shimadzu RF-5301PC, Kyoto, Japan) at 37 °C. After initial baseline of fluorescence intensity was collected, ferrous ammonium sulfate (FAS, 40 µM, final concentration, #203505, Sigma) was added to the cuvette. The quenching of Calcein fluorescence was recorded in every 5 min for 30 min.

### 4.9. Statistical Analysis

All data were presented as mean ± SD. Statistical analyses were performed using GraphPad Prism 7.0. The differences between the means were determined by One-Way or Two-Way ANOVA followed by a Tukey post-hoc test for multiple comparisons. The figures were drawn by GraphPad Prism 7.0 software. A probability value of *p* < 0.05 was taken to be statistically significant.

## 5. Conclusions

In summary, we demonstrate for the first time that iron can exacerbate the loss of dopaminergic neurons in the substantia nigra of PD model rats caused by 6-OHDA and induce the activation of LRRK2, promoting the expression of pro-apoptotic factors and ROS production in SH-SY5Y cells. The activation of LRRK2 caused by 6-OHDA is mediated mainly by iron. Interestingly, activated LRRK2 can in turn enhance ferrous iron uptake, leading to an increase in the level of total iron in SH-SY5Y cells. Our study provides important evidence that there is an interplay between iron and LRRK2 in dopaminergic cells. These findings extend the valuable insights gained from exploring mechanisms underlying PD occurrence.

## Figures and Tables

**Figure 1 ijms-24-03709-f001:**
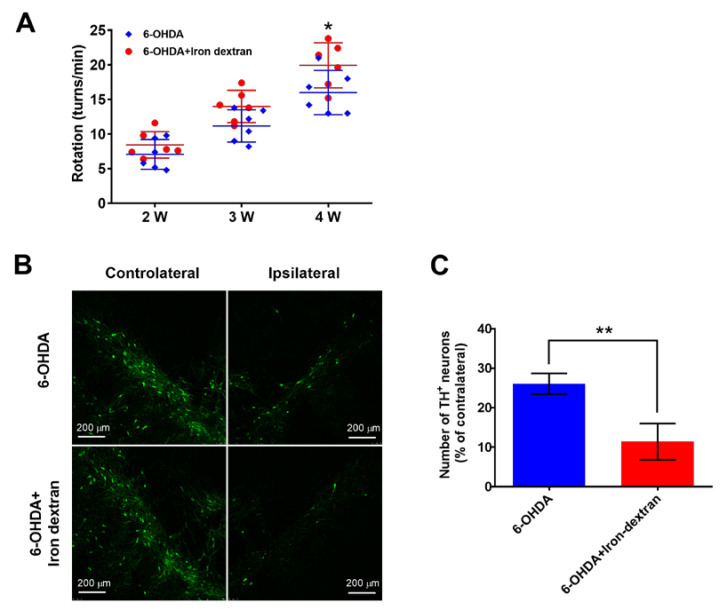
Iron dextran exacerbates the loss of dopaminergic neurons in the substantia nigra of PD rats. Rats were administered 6-OHDA into the stratum with or without iron dextran into the MFB. (**A**) Rotational behavior test on PD model rats. Rotational behavior of rats was induced by apomorphine (0.5 mg/kg, s.c.) two, three, and four weeks after 6-OHDA administration (n = 6). * *p* < 0.05 versus 6-OHDA after four weeks. (**B**) Representative images of dopaminergic neurons in the substantia nigra of rats. Brain slices containing substantia nigra pars compacta after four weeks were immunostained by anti-TH antibody, left: Controlateral side, right: Ipsilateral side. (**C**) Quantification of dopaminergic neurons (n = 6). ** *p* < 0.01. Data were obtained from three independent experiments.

**Figure 2 ijms-24-03709-f002:**
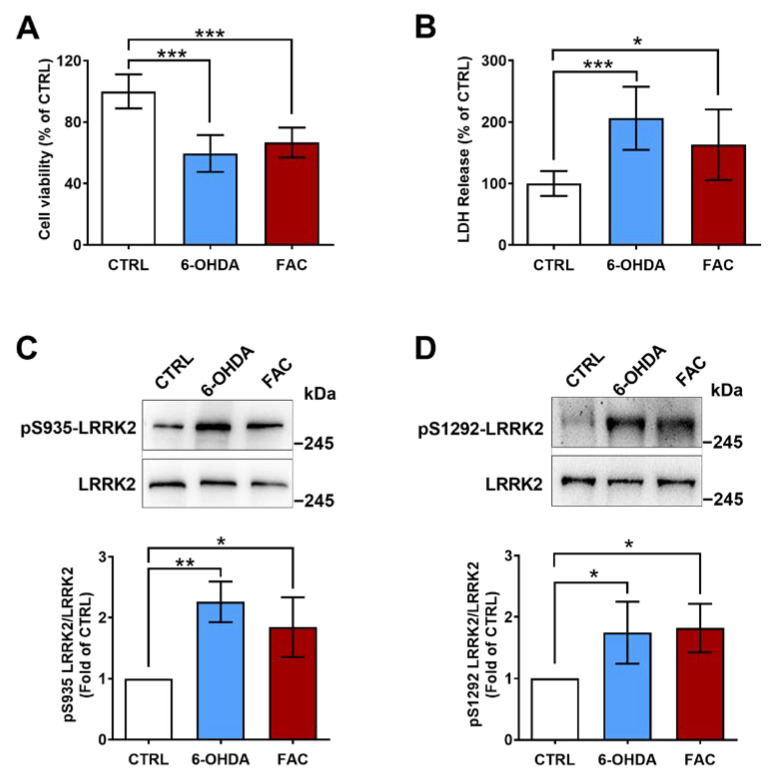
6-OHDA and FAC decrease cell viability and induce LRRK2 activation. SH-SY5Y cells were treated with 100 μM of 6-OHDA or 5 μg/mL of FAC for 24 h. (**A**) Cell viability by MTT assay (n = 12). (**B**) Cell viability by LDH assay. After 6-OHDA and FAC treatment, the cellular medium was collected and analyzed by LDH assay (n = 10). (**C**) The level of pS935-LRRK2. (**D**) The level of pS1292-LRRK2. Upper: representative Western blot images of pS935-LRRK2 or pS1292-LRRK2, lower: quantification of the levels of pS935-LRRK2 or pS1292-LRRK2 in SH-SY5Y cells (n = 3). * *p* < 0.05, ** *p* < 0.01, *** *p* < 0.001.

**Figure 3 ijms-24-03709-f003:**
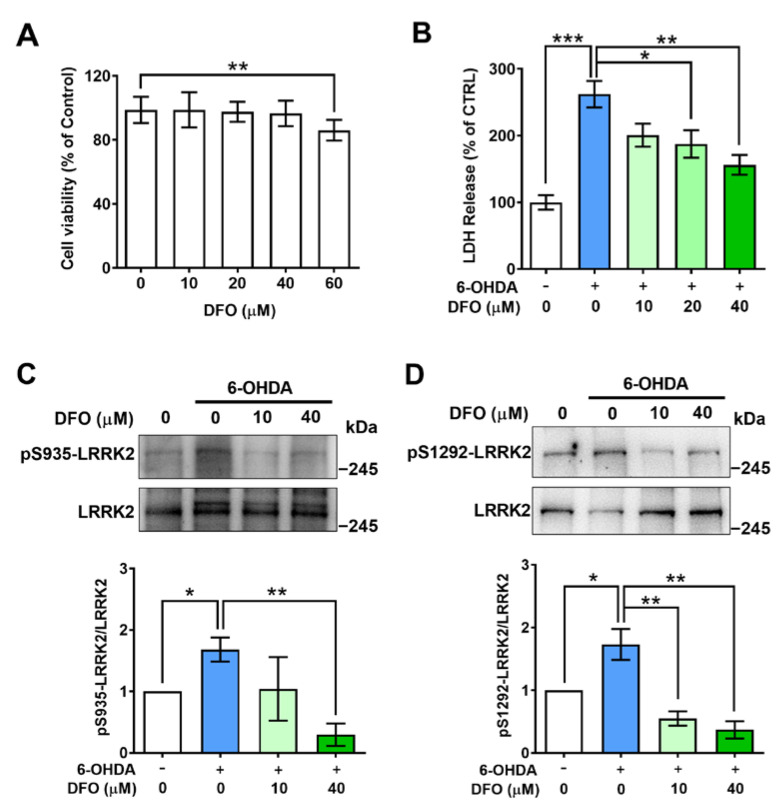
Desferrioxamine inhibits activation of LRRK2 and cell damage caused by 6-OHDA. SH-SY5Y cells were treated with 0, 10, 20, 40, or 60 μM of DFO for 24 h. (**A**) The cell viability was analyzed by the MTT assay (n = 10). (**B**) Effect of DFO on the survival of SH-SY5Y cells by LDH assay. SH-SY5Y cells were incubated with 0, 10, 20, or 40 μM DFO for 24 h in the presence of 100 μM 6-OHDA. (**C**) The level of pS935-LRRK2. (**D**) The level of pS1292-LRRK2. SH-SY5Y cells were incubated with 0, 10, or 40 μM DFO for 24 h in the presence of 100 μM 6-OHDA. Upper: representative Western blot images of pS935-LRRK2 or pS1292-LRRK2, lower: quantification of the levels of pS935-LRRK2 or pS1292-LRRK2 in SH-SY5Y cells (n = 3). * *p* < 0.05, ** *p* < 0.01, *** *p* < 0.001.

**Figure 4 ijms-24-03709-f004:**
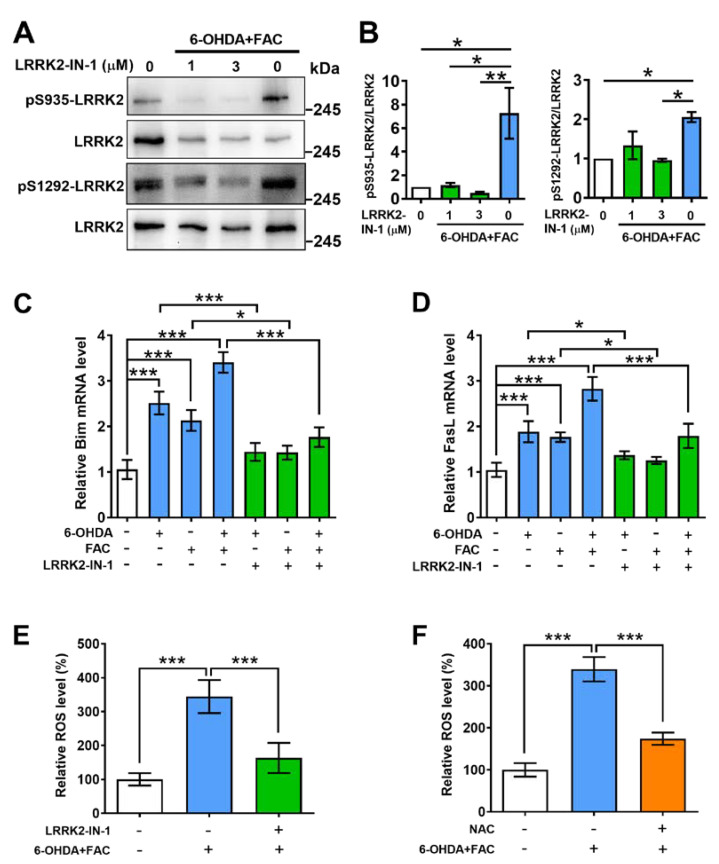
LRRK2-IN-1 inhibits phosphorylation of LRRK2, expression of pro-apoptotic molecules, and production of ROS in SH-SY5Y cells. SH-SY5Y cells were incubated with the inhibitor of LRRK2 activation, LRRK2-IN-1, at 0, 1, or 3 μM for 1 h before 6-OHDA plus FAC treatment (6-OHDA+FAC) for 24 h. (**A**) Representative Western blot images of pS935-LRRK2 or pS1292-LRRK2. (**B**) Quantification of the levels of pS935-LRRK2 or pS1292-LRRK2 in SH-SY5Y cells (n = 3). (**C**,**D**) Expression of Bim and FasL mRNA in SH-SY5Y cells. SH-SY5Y cells were incubated with the inhibitor LRRK2-IN-1 at 3 μM for 1 h before 6-OHDA, FAC, or 6-OHDA+FAC treatment for 24 h (n = 3). (**E**,**F**) The relative ROS level in SH-SY5Y cells was determined by the DCFH-DA assay. SH-SY5Y cells were incubated with LRRK2-IN-1 at 3 μM or N-acetylcycsteine (NAC) at 5 mM for 1 h before 6-OHDA+FAC treatment for 24 h, after which intracellular ROS was measured by the DCFH-DA assay (n = 3). * *p* < 0.05, ** *p* < 0.01, *** *p* < 0.001.

**Figure 5 ijms-24-03709-f005:**
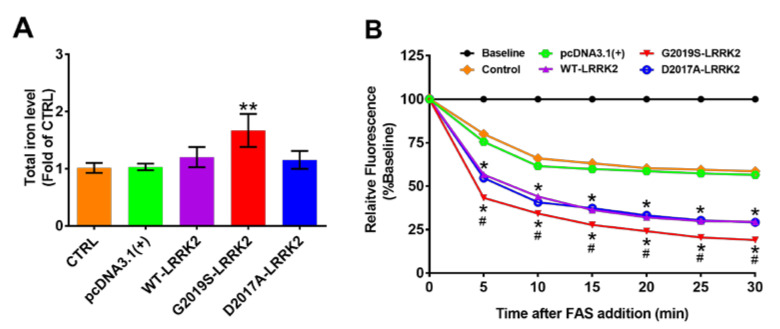
G2019S-LRRK2 increases total iron content and ferrous iron uptake. (**A**) The intracellular total iron content was measured by reacting with Ferene S after ferric reduction to ferrous (n = 3). ** *p* < 0.001 versus the pcDNA3.1(+) vector. (**B**) The ferrous iron uptake was investigated by Calcein-AM. After the initial baseline of fluorescence intensity was collected at 488 nm for excitation and 525 nm for emission wavelengths, ferrous ammonium sulfate (FAS; 40 µM of final concentration) was added to the cuvette. The quenching of calcein fluorescence was recorded every 5 min for 30 min (n = 3). * *p* < 0.001 versus the pcDNA3.1(+) vector, ^#^
*p* < 0.001 versus WT-LRRK2 at each time.

## Data Availability

All data produced for this manuscript are available from the lead contact (wuxmsu@ntu.edu.cn) upon reasonable request. Informed consent was obtained from all subjects involved in the study.

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
