# Peer review of "The Relationship between Iron and LRRK2 in a 6-OHDA-Induced Parkinson’s Disease Model"

_ijms, 2023, doi:10.3390/ijms24043709_

Round 1

Reviewer 1 Report

The authors showed an important interplay between iron and LRRK2 in dopaminergic cells and consequently PD occurrence. However, the authors should describe the limitations of the present study and what other experiments should be carried out to support their important conclusions.

Author Response

Thank you very much for your suggenstions. We have added the limitations of the present study and what other experiments should be carried out in the future study in disccusion section.

Reviewer 2 Report

This is an interesting study discussing the correlation between iron and LRRK2 in PD models, however, the available experimental data are relatively preliminary. I have several questions as follows.

Figure 1, more supplements can be made in the section of in vivo experiments. For example, Perform TH staining of serial sections include different substantia nigra tissue levels; Perform double staining for LRRK2/pLRRK2 and TH; Perform iron/apoptosis assay.

The authors used SH-SY5Y for cell experiments rather than dopaminergic neurons, then, it remains unclear how LRRK2 and iron are associated within dopaminergic neurons as well as in other neurons.

The authors measured the protein expression level of LRRK2/pLRRK2, but beta-action is still an important internal control which is missing.

Round 2

Reviewer 2 Report

I have no further comments.

Author Response

Thank you very much.